# Solid-phase hetero epitaxial growth of α-phase formamidinium perovskite

Jin-Wook Lee[1,2,7 ✉], Shaun Tan[1,7], Tae-Hee Han[1,3,7], Rui Wang[1], Lizhi Zhang[4], Changwon Park[4], Mina Yoon[4], Chungseok Choi[1], Mingjie Xu[5], Michael E. Liao[1], Sung-Joon Lee[1], Selbi Nuryyeva[1], Chenhui Zhu[6], Kenny Huynh[1], Mark S. Goorsky[1], Yu Huang[1], Xiaoqing Pan[5] & Yang Yang[1 ✉]

Conventional epitaxy of semiconductor films requires a compatible single crystalline substrate and precisely controlled growth conditions, which limit the price competitiveness and versatility of the process. We demonstrate substrate-tolerant nano-heteroepitaxy (NHE) of high-quality formamidinium-lead-tri-iodide (FAPbI$_3$) perovskite films. The layered perovskite templates the solid-state phase conversion of FAPbI$_3$ from its hexagonal non-perovskite phase to the cubic perovskite polymorph, where the growth kinetics are controlled by a synergistic effect between strain and entropy. The slow heteroepitaxial crystal growth enlarged the perovskite crystals by 10-fold with a reduced defect density and strong preferred orientation. This NHE is readily applicable to various substrates used for devices. The proof-of-concept solar cell and light-emitting diode devices based on the NHE-FAPbI$_3$ showed efficiencies and stabilities superior to those of devices fabricated without NHE.

[1] Department of Materials Science and Engineering, California NanoSystems Institute, University of California, Los Angeles, CA 90095, USA. [2] Department of Nanoengineering, SKKU Advanced Institute of Nanotechnology (SAINT), Sungkyunkwan University, Suwon 16419, Republic of Korea. [3] Division of Materials Science and Engineering, Hanyang University, Seoul 04763, Republic of Korea. [4] Department of Physics and Astronomy, University of Tennessee, Knoxville, TN 37996, USA. [5] Department of Materials Science and Engineering, Irvine Materials Research Institute, University of California, Irvine, CA 92697, USA. [6] Advanced Light Source, Lawrence Berkeley National Laboratory, Berkeley, CA 94704, USA. [7] These authors contributed equally: Jin-Wook Lee, Shaun Tan, Tae-Hee Han. ✉email: jw.lee@skku.edu; yangy@ucla.edu

Epitaxial growth is one of the most powerful ways to fabricate semiconducting thin films with low defect density and a desired orientation[1]. The technique has been widely adopted to form semiconducting thin films for optoelectronic applications for which the defect density should be minimized to promote carrier transport and thus device performance. Highly crystalline semiconducting films including silicon, gallium nitride, gallium arsenide, and two-dimensional materials have been successfully grown via epitaxial growth to be used in solar cells, light-emitting diodes (LEDs) and memory devices[2–6]. However, conventional epitaxy requires a compatible single crystalline substrate, and the growth process is performed in extremely controlled environments such as ultra-high vacuum and temperature. Often, the grown materials should be transferred from the templating substrate onto a desired substrate for device fabrication by delicate lift-off processes. These limit not only the price competitiveness and scalability, but also versatility of the process for various materials and devices.

For instance, the epitaxial growth of metal halide perovskite thin films for used in devices has not yet been achievable because of (i) the absence of compatible substrates to template the growth of the perovskite film, and (ii) the difficulty of inducing controlled nucleation and growth at the surface of a templating substrate. For the former, typical transparent oxide substrates and other charge transporting bottom contact layers used for devices cannot direct the growth of the perovskite film. Single crystalline chunks of halide perovskites and epitaxially grown thin films on single crystal substrate have been demonstrated[7–11], but thickness-controlled layer transfer of the grown materials onto a desired substrate has not been successful. For the latter, inherently fast reaction kinetics during the typical solution process causes numerous nuclei to form in the bulk solution, resulting in the growth of randomly oriented fine grains with a high density of defects. In light of these limitations, huge research efforts have been devoted to control the nucleation and growth kinetics[12,13], but accurate control over the kinetics in solution medium is still challenging.

Recently, strategies to incorporate layered perovskites into 3D perovskites have been extensively studied. Owing to their enhanced stability relative to 3D perovskites and favorable band alignment, layered perovskites effectively passivate the grain boundaries and surface of 3D perovskites, resulting in enhanced performance and stability of the perovskite solar cells and LEDs[14–17]. We present here a route to induce kinetic-controlled epitaxial crystal growth of formamidinium lead tri-iodide (FAPbI$_3$) perovskite thin films by using layered perovskite templates. The local epitaxial growth of the FAPbI$_3$ perovskite crystal was observed during its solid-state phase transformation from the hexagonal non-perovskite FAPbI$_3$ when it is heterostructured with layered perovskite, whereby the growth kinetics was dependent on strain energy originating from the hetero-interface. Our first-principles calculations revealed a mechanism to engineer the conversion energy barrier between the cubic and hexagonal phases by a synergistic effect between strain and entropy. The slow heteroepitaxy enabled the growth of tenfold enlarged FAPbI$_3$ perovskite crystals with a reduced defect density and strong preferred orientation. This nano-heteroepitaxy (NHE) is applicable to various substrates used for devices. The proof-of-concept solar cell and LED devices based on NHE–FAPbI$_3$ showed efficiencies and stabilities superior to those of devices fabricated without NHE.

## Results

**Phase conversion kinetics**. FAPbI$_3$ crystallizes as its equilibrium hexagonal non-perovskite phase (yellow color, δ-phase) at room temperature, and is transformed into its cubic perovskite polymorph (black color, α-phase) upon annealing at 150 °C through a soild-state phase conversion process[18,19]. In polycrystalline films, grain boundaries are at a higher energy state due to the additional contribution of surface (or interfacial) energy. Consequently, the grain boundaries act as heterogeneous nucleation sites where phase transition is first triggered. We introduced a hetero-interface at the grain boundaries by incorporating small amounts of layered perovskites into the precursor solution[14,15]. We varied both the amount and type of the layered perovskite to observe the effects of the hetero-interface (hereafter we define, control: bare FAPbI$_3$, 1P: FAPbI$_3$ with 1.67 mol% of PEA$_2$PbI$_4$, 3P: FAPbI$_3$ with 3.33 mol% of PEA$_2$PbI$_4$, and 3F: FAPbI$_3$ with 3.33 mol% of FPEA$_2$PbI$_4$). We expected the hetero-interface area to increase in the order control < 1P < 3P < 3F (please refer to Supplementary Note 1 and Supplementary Figs. 1 and 2 for discussion on the differences between PEA$_2$PbI$_4$ and FPEA$_4$PbI$_4$).

The layered perovskite/FAPbI$_3$ hetero-structured films were formed by spin-coating the precursor solutions, and the photographs of the films as a function of annealing time are presented in Fig. 1a. As-spun films were semitransparent yellowish irrespective of the added layered perovskite, indicating the formation of the hexagonal δ-phase. Upon annealing at 150 °C, the color of the control film rapidly converted to black within a minute, whereas the color change was retarded with incorporation of the layered perovskites; 1P, 3P, and 3F took approximately 2, 6, and 8 min, respectively. The slower color change is indicative of a slower phase conversion process. In Fig. 1b, the phase conversion process was monitored by in situ grazing incident wide angle X-ray scattering (GIWAXS) measurement of the corresponding films at 150 °C (under helium atmosphere on silicon substrates). Before heating at 150 °C, all films showed a characteristic scattering peak around $q = 0.842\,\text{Å}^{-1}$, corresponding to the (010) orientation peak of the δ-FAPbI$_3$ phase. As the films were annealed at 150 °C, the peak disappeared with simultaneous occurrence of a new peak around $q = 0.996\,\text{Å}^{-1}$, attributed to the (001) orientation peak of the α-FAPbI$_3$ phase. The δ(010) peak of the control film almost abruptly disappeared after heating for 1 min with simultaneous appearance of the α(001) peak, and afterwards the α(001) peak maintained its initial shape and intensity. On the contrary, the δ(010) peak of the films with layered perovskites was retained for longer during the heat treatment and gradually converted to the α(001) peak, indicating slower phase conversion with the layered perovskites. To analyze the phase conversion kinetics, we plotted isothermal transformation diagrams as in Fig. 1c. The relative peak intensity (integrated area under the peaks) of α(001) to δ(010) was calculated from the in situ GIWAXS patterns as a function of annealing time (upper panel in Fig. 1c). A delayed phase transformation with incorporation of the layered perovskites from control, to 1P, to 3P, and to 3F can be clearly seen. To quantitatively estimate the transformation kinetics, we fitted the phase conversion profiles using the simplified Johnson–Mehl–Avrami equation which describes the relationship between the fraction of transformed phase and the elapsed time (please refer to Supplementary Note 2 for details). The extracted k values (Supplementary Table 1) are seen to decrease from the control, 1P, 3P, to 3F films, suggesting that the nucleation and/or growth rates of the cubic phase were significantly retarded by the incorporation of the layered perovskites. We also confirmed that the phase transformation kinetics followed the same tendency for films on SnO$_2$-coated indium tin oxide (ITO) substrates depending on the film composition by measuring the X-ray diffraction (XRD) patterns of the films with different heating times (Supplementary Fig. 3 and the lower panel in Fig. 1c). In addition to the different kinetics, the GIWAXS peak shapes also evolved differently for the films with layered perovskites as compared to

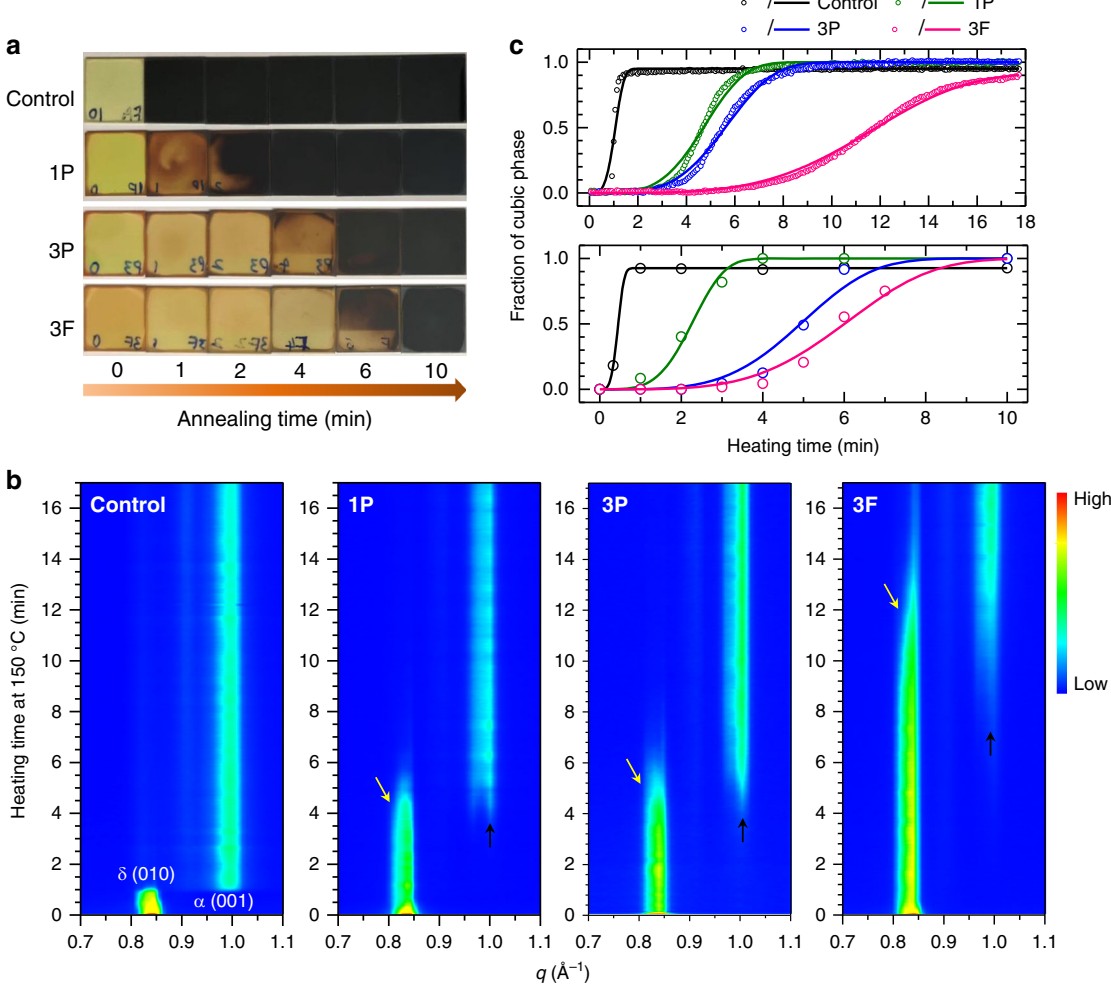

**Fig. 1 Phase conversion kinetics of formamidinium lead tri-iodide (FAPbI₃) perovskite with a hetero-interface at the grain boundaries. a** Photographs of FAPbI₃ films on SnO₂-coated ITO substrates with different annealing times at 150 °C. Control: bare FAPbI₃, 1P: FAPbI₃ with 1.67 mol% PEA₂PbI₄, 3P: FAPbI₃ with 3.33 mol% PEA₂PbI₄, and 3F: FAPbI₃ with 3.33 mol% FPEA₂PbI₄. **b** In situ grazing incident wide angle X-ray scattering (GIWAXS) measurements of corresponding films deposited on silicon wafer substrates. **c** Isothermal transformation diagrams showing the evolution of the α-FAPbI₃ phase proportion in the films as a function of annealing time at 150 °C on silicon wafers (upper panel) and on SnO₂-coated ITO substrates (lower panel).

the control film. The portion of the δ(010) peak with relatively lower $q$ (higher $d$ spacing, yellow arrows in Fig. 1b) disappeared faster, whereas the portion of the α(001) peak with relatively higher $q$ (lower $d$ spacing, black arrows in Fig. 1b) emerged faster. This feature became more pronounced as the phase conversion process retarded further, which implies the presence of strained phases and a consequent change in energetics and kinetics of the phase conversion process. We speculated that the introduction of the hetero-interface between the FAPbI₃ and layered perovskites was probably associated with the observed strain and was responsible for retarding the phase conversion process.

**Microstructure analysis and first-principles modeling**. To observe the microstructure evolution during the phase conversion, in situ transmission electron microscopy (TEM) measurement was conducted on the 3F film at an elevated temperature (Fig. 2a–d). Upon heating the sample, nuclei of the cubic phase (bright spots indicated by arrows) formed simultaneously at both the bulk and boundaries of the grain (Fig. 2a). However, the bulk nuclei were redissolved within a few minutes (Fig. 2b), which is probably due to the relatively higher free energy barrier for stable nucleation. On the contrary, the grain boundary nucleus was

stable and slowly grew towards the grain interior (Fig. 2c). As the growth proceeded, we observed that the distinct grain boundary present before the phase conversion gradually became less distinguishable, and thus we speculated that the adjacent grains possibly merged during the phase transition (Fig. 2d). In addition, the layered perovskites at the grain boundary region grew and reoriented/diffused during the phase conversion process (Supplementary Fig. 4a–d). As a result, we observed aggregated layered perovskites along with blurred grain boundaries of the FAPbI₃ film from the cross-sectional TEM images of the sample after the phase conversion process (Supplementary Fig. 4e, f). We further noticed that the lattice of the as-formed α-FAPbI₃ seemed to have aligned with the aggregated layered perovskites (Supplementary Fig. 4g, h). *Note*: Since the in-situ measurement was carried out under vacuum on a sample with thickness <50 nm, the resulting microscopic morphology (grain size and layered perovskite distribution) may be different from that processed under ambient conditions. High-resolution TEM images of the 3F film were further analyzed to investigate the microstructure of the hetero-interface between FAPbI₃ and the layered perovskite (Fig. 2e–j and Supplementary Fig. 5). From the scanning TEM (STEM) images in Supplementary Fig. 5a, b, we identified the layered perovskite at the grain boundaries and surface of the

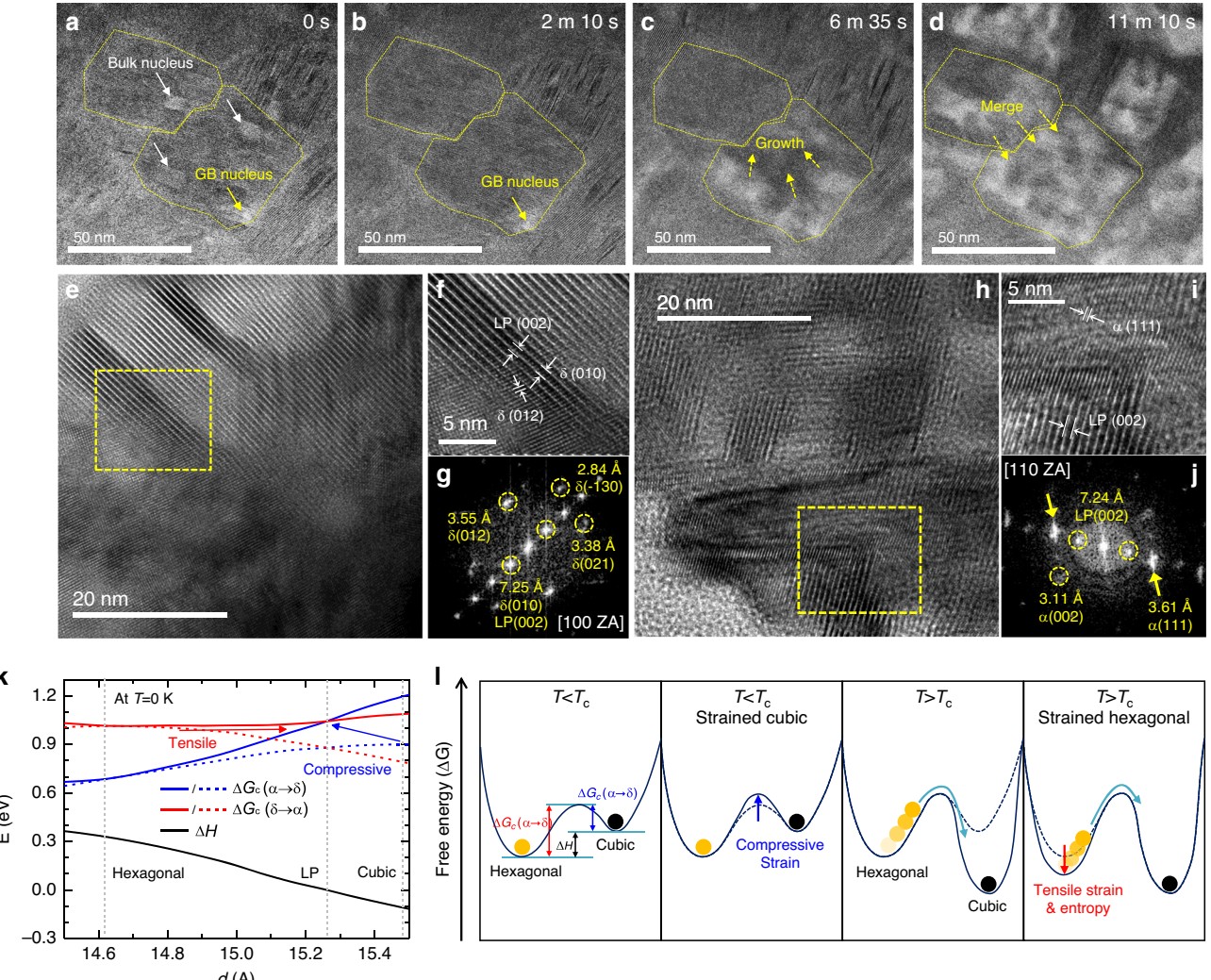

**Fig. 2 Transmission electron microscopy (TEM) images and density functional theory (DFT) calculations. a–d** In situ TEM images of the δ-FAPbI₃ film annealed at the phase conversion temperature for different times. The relatively bright spots are the cubic phase nuclei transformed from the hexagonal phase. High-resolution TEM images of FAPbI₃ films incorporated with 3.33 mol% FPEA₂PbI₄ **e**, **f**, before and **h**, **i** after the phase conversion process. **f**, **i** Magnified TEM images of the regions highlighted with yellow dashed boxes in (**e**) and (**h**), respectively. **g**, **j** Corresponding fast Fourier transform (FFT) analysis of images (**f**) and (**i**), respectively. Figure 2e–j are from samples prepared by scratching off the films from the substrate. **k** DFT-calculated free energy barriers (ΔGₛs) for phase conversion from cubic to hexagonal (α → δ), from hexagonal to cubic (δ → α), and formation enthalpy of the cubic phase (ΔH) with respect to that of the hexagonal phase. Dashed lines and solid lines indicate ΔGₛ without and with strain, respectively. The interlayer spacing of the strain-free hexagonal phase, layered perovskite and cubic phase are indicated with gray colored dashed vertical lines (d-spacings of DFT optimized structures). **l** Schematic free energy diagrams for the hexagonal and cubic phased FAPbI₃ at different temperatures and strain condition. T is temperature and Tₛ is the temperature for phase conversion.

FAPbI₃ grains before and after the heat-treatment with its characteristic interplanar spacing of around 7.25 Å corresponding to the (002) plane[20,21]. Figure 2f shows the magnified interface region between the layered perovskite and the δ-FAPbI₃ phase before the phase conversion process. From fast Fourier transform (FFT) analysis of the image in Fig. 2g, we found that the (002) plane of the layered perovskite is aligned with the (010) plane of the δ-FAPbI₃ phase. The measured interplanar distance of the δ-FAPbI₃ (7.25 Å) at the interface deviated from that measured from their bulk counterparts; 7.11 Å for (010) of δ-FAPbI₃, Fig. 2g and Supplementary Fig. 6a–c. The larger interplanar spacing of the δ-FAPbI₃ indicates the presence of tensile strain at the interface with the layered perovskite. From the TEM images and corresponding FFT analysis after the phase conversion process (Fig. 2i, j), a semi-coherent interface between the layered perovskite and the α-FAPbI₃ was observed. Despite

of the semi-coherent interface, the (111) plane of the grown α-FAPbI₃ orientationally aligned with the (002) plane of the layered perovskite, which implies that heteroepitaxial growth of the α-FAPbI₃ crystals occurred during the phase conversion process. The measured interplanar spacing of 3.61 Å for the (111) plane of the α-FAPbI₃ at the interface region was slightly smaller than that measured from its bulk counterpart (3.65–3.71 Å, Fig. 2j and Supplementary Fig. 6a–f), which implies the existence of compressive strain applied to the interfacial α-FAPbI₃ phase. From this observation, we assumed that the δ-FAPbI₃ phase (with its larger interplanar spacing) being converted to α-FAPbI₃ (with its smaller interplanar spacing) at the initial stage of the phase conversion (as shown in the in situ GIWAXS measurements in Fig. 1b with layered perovskite) was probably the FAPbI₃ at the interface with the layered perovskite. This indicates that the phase transition initiates at the

hetero-interface, and the formed nuclei might grow towards the bulk of the grain.

To elucidate the origin of the different phase conversion kinetics, we performed first-principles density functional theory (DFT) calculations based on the observed strain and interfacial alignments as determined by the TEM studies (Supplementary Fig. 7). We initially postulated that two factors may have contributed to the change in phase conversion energetics: (i) strain itself which can be directly responsible for increasing the phase conversion nucleation barrier as is well-known from conventional nucleation theory[22], and (ii) change in entropy induced by the presence of strain, which was previously reported to be a crucial factor affecting the energetics of FAPbI$_3$[19]. The phase conversion energetics without strain is presented in Supplementary Fig. 8. Firstly, to investigate the effect of strain, the energy barriers ($\Delta G_c$) between the cubic and hexagonal phases with the lattice constants fixed to the given interlayer spacings are calculated as in Fig. 2k. Solid lines indicate the energy barriers with strain applied to the transition states, whereas the dotted lines are the barriers with relaxed (unstrained) transition states (interlayer spacings of only the hexagonal and cubic phases are constrained to the given interlayer spacings). Overall, the energy barrier increases when strain is applied to the transition state as expected from classical nucleation theory[22]. As the interlayer spacing decreases toward the 2D spacing, the compressive strain increases the cubic to hexagonal transformation barrier height by ~0.15 eV in comparison to the value for the fully relaxed lattice (blue arrows in Fig. 2k and second panel in Fig. 2l), which will be beneficial for the thermodynamic stability of the cubic phase at $T < T_c$ ($T_c$ is the critical temperature for phase conversion), correlating with previous experimental observations[9]. In the case of the phase transition from the hexagonal to cubic phase ($\delta \rightarrow \alpha$), strain seems to less affect the $\Delta G_c$ value. However, we found that the contribution of entropy becomes more dominant. The activated rotation of the FA molecule at elevated temperatures contributes to the entropy of the system, which is known to be more significant in isotropic cubic structures and thus stabilizes the cubic phase over the hexagonal phase at a finite temperature[19]. The entropy contribution ($S_{rot}$) of a freely rotating FA$^+$ cation becomes more significant with an increasing temperature ($T$) as given by

$$S_{rot} = \frac{3}{2}k_B\left[1 + \ln\left(0.4786 k_B T \sqrt[3]{I_1 I_2 I_3}\right)\right],$$

where $k_B$ and $I_i$ are the Boltzmann constant and principal moments of inertia, respectively. The principal moments of inertia ($I_i$) of the FA$^+$ cation are 8.586, 48.851, and 57.436 uÅ$^2$, where the unified atomic mass unit, u, is $1.6605 \times 10^{-27}$ kg[19]. The differences in the Gibbs free energy between the two phases are about $-0.27$ eV at $T = 300$ K and $-0.47$ eV at $T = 500$ K. As the free energy linearly decreases with $T$ for the cubic phase, the relative stability between the two phases inverted above $T_c$ while its contribution is negligible for the hexagonal phase (third panel in Fig. 2l)[19]. On the other hand, the rotational entropy becomes a key contributor in stabilizing the strained hexagonal phase. Under the tensile strain, the interlayer spacing of the hexagonal phase increases significantly, and the activated isotropic rotation of the FA$^+$ cation also becomes accessible for the hexagonal phase. We confirmed a more than 50% decrease in the rotational barrier induced by the tensile strain, while the rotational barrier for the compressed cubic phase is not much affected (Supplementary Fig. 9). Consequently, the tensile strain applied to the hexagonal phase effectively increases the stability of the hexagonal phase at a finite temperature and contributes to slowing down the phase transformation process to the cubic phase (red arrow in the last panel in Fig. 2l).

**Effect of the nano heteroepitaxy on film quality**. The effect of the NHE on film quality was investigated by comparing the XRD patterns of the films before and after the phase conversion process. In Fig. 3a, all the as-fabricated films showed characteristic peaks originating from the δ-FAPbI$_3$ phase with no significant difference in peak intensity, indicating that the added layered perovskites do not significantly alter the crystallization process of the hexagonal phase from the precursor solutions. On the contrary, the XRD patterns of the films after the phase conversion process were considerably different depending on the composition (Fig. 3b). The intensity of the (001) and (002) peaks originating from the α-FAPbI$_3$ phase were systematically enhanced (control < 1P < 3P < 3F) as the phase conversion became slower, which is also obvious in the high-resolution XRD patterns in Fig. 3c. The enhancement of selective peaks of a specific orientation is indicative of an enhanced preferred orientation of the films. We confirmed the systematic enhancement of the preferred orientation of the films from pole figure measurements along the (001) orientation (Fig. 3d–g). The more localized signal intensity at the center of the circle can be interpreted as a higher degree of preferred orientation of the films normal to the substrate. The degree of the (001) preferred orientation measured from the pole figures was well-correlated with the XRD patterns in Fig. 3b, c (control < 1P < 3P < 3F). The stronger preferred orientation indicates that heteroepitaxial growth of the α-FAPbI$_3$ phase became increasingly dominant with a larger hetero-interface area.

Closer inspection of the high resolution XRD patterns revealed that the full-width-half-maximums of the normalized (002) peaks decreased as the peaks were gradually shifted toward higher angles in the same order, which indicates an incrementally increasing crystallite size and compressive strain (inset of Fig. 3c). The crystallite size and lattice strain of the δ-FAPbI$_3$ films were further extracted from the full XRD patterns by using the Williamson–Hall method (Supplementary Note 3, Supplementary Fig. 10, Supplementary Tables 2 and 3)[23,24]. Since the Williamson–Hall method is limited to crystallite sizes below 100 nm, for the α-FAPbI$_3$ crystallite sizes, we extracted the grain sizes of the films from the atomic force microscopy images in Supplementary Fig. 11a–d, and their distributions are shown in Supplementary Fig. 12. The results are summarized in Fig. 3h. The calculated strain of the films before phase conversion was closely correlated with the phase conversion kinetics, which confirms that the strain energy and resulting entropy change associated with the hetero-interface were likely responsible for the retarded phase conversion kinetics. The crystallite sizes before phase conversion were comparable irrespective of composition. Notably, the crystallite sizes were significantly enlarged as the phase conversion became slower (from 43.5 to 160.4 nm for control, from 43.7 to 417.9 nm for 1P, from 40.6 to 520.8 nm for 3P, and from 65.4 to 675.0 nm for 3F). Based on the observations, we propose the following mechanism for the NHE (Supplementary Fig. 13). (i) The added layered perovskite crystallizes at the grain boundaries with high surface energy, which would have otherwise triggered the initial nuclei formation. (ii) The strain and resulting entropy change induced by the hetero-interface retards the phase transition and local epitaxial growth proceeds from the interface. (iii) Owing to active reconstruction processes involving breakage and reformation of bonds during the phase transition[19,25], merger of the FAPbI$_3$ grains and layered perovskite sheets can be thermally activated. (iv) Due to preferential orientation of the aggregated layered perovskite[26], the FAPbI$_3$ grains also orient with respect to the substrate. Our first-principles surface energy calculations support the suggested mechanism, where the surface energies of the δ(010) and α(111) facets are relatively higher than those of other facets

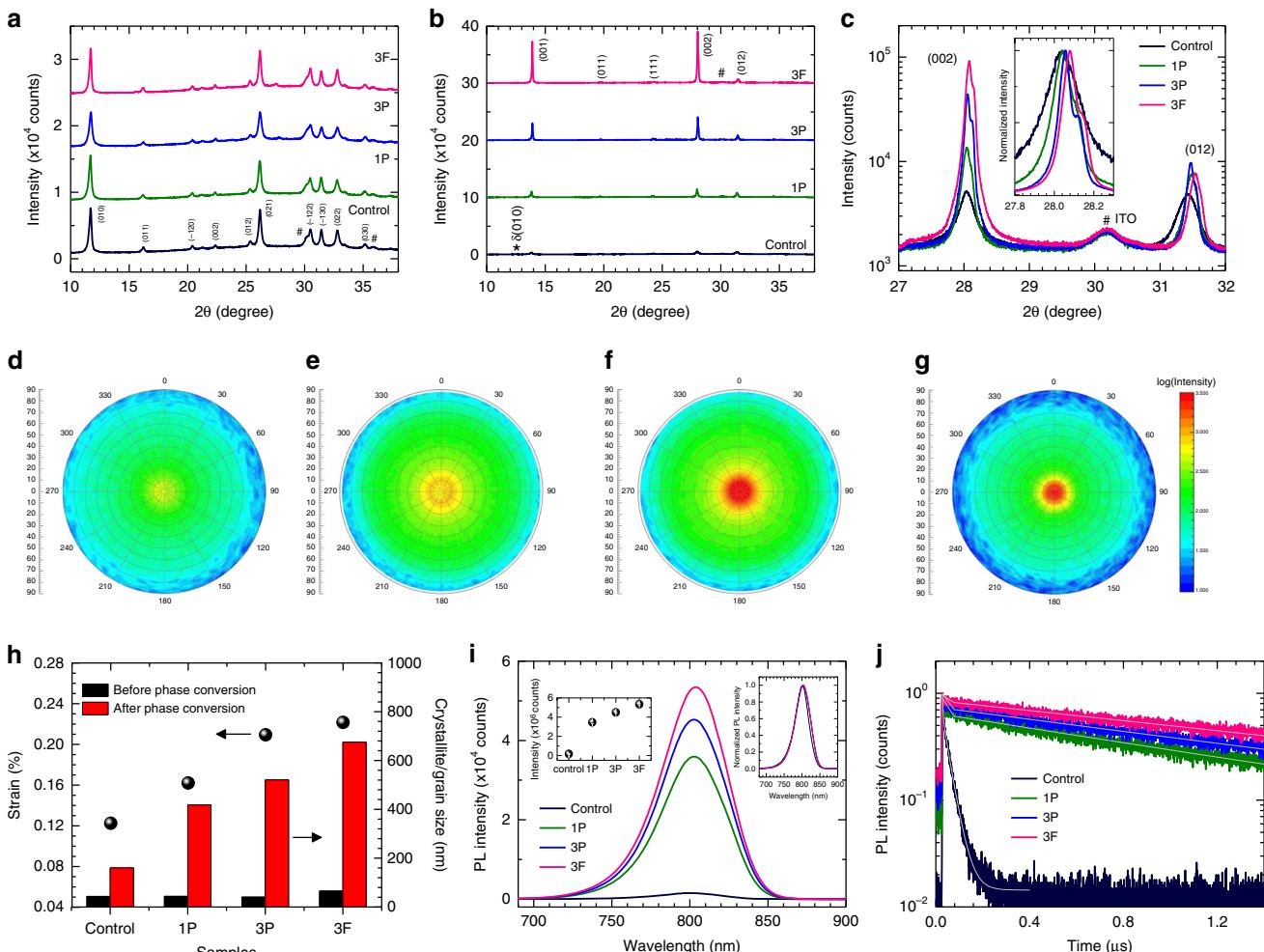

**Fig. 3 Crystallographic analyses before and after the phase conversion.** X-ray diffraction (XRD) patterns of the films **a** before and **b** after the phase conversion process. **c** High-resolution X-ray diffraction (XRD) patterns of the films after the phase conversion process. Inset shows the normalized (002) orientation peaks. **d–g** XRD pole figure measurements along the (001) orientation of the **d** control, **e** 1P, **f** 3P, and **g** 3F films. **h** Composition-dependent strain before phase conversion and crystallite size of the films before and after the phase conversion process. The strain was calculated using the Williamson-Hall method for the $\delta$-FAPbI$_3$ films. The $\delta$-FAPbI$_3$ crystallite sizes were calculated by the Williamson–Hall method, while the $\alpha$-FAPbI$_3$ crystallite sizes were extracted from the AFM images. **i** Steady-state photoluminescence (PL) and **j** time-resolved PL decay profile measurements of the corresponding films. Insets of **i** show peak intensity (left) and normalized PL spectra (right).

(Supplementary Fig. 14), which might be the origin of the preferential alignment of the layered perovskites at the crystal facets.

Steady-state photoluminescence (PL) spectra of the films are presented in Fig. 3i. The near-identical peak positions irrespective of film composition was correlated with the identical absorption onsets (inset of Fig. 3i and Supplementary Fig. 15), which demonstrates that the bandgap of the films is unchanged from that of bulk FAPbI$_3$. The films with layered perovskites showed enhanced absorptions over the whole wavelength region as compared to the control film, which is probably due to the higher phase purity as observed in our previous study[27]. The peak PL intensity of $0.16 \times 10^6$ for the control film was significantly enhanced with NHE to $3.46 \times 10^6$, $4.49 \times 10^6$, and $5.34 \times 10^6$ for the 1P, 3P, and 3F films, respectively. Time-resolved PL decay profiles of the films are included in Fig. 3j to elucidate the enhanced PL intensities. The measured data were fitted to a bi-exponential decay model, and the fitted parameters are summarized in Supplementary Table 4. All films showed two decay regimes: a relatively faster initial decay ($\tau_1$ ~3 ns), followed by a much slower decay ($\tau_2$ » 10 ns). The faster decay component

is generally assigned to charge carrier trapping caused by deep trap states originating from structural defects, whereas the slower decay is attributed to bimolecular radiative recombination of the free carriers[28,29]. It should be noted that the fast decay proportion ($A_1$) decreased from 39.1% for the control film to 38.2%, 27.8%, and 13.4% for the 1P, 3P, and 3F films, respectively. The decreased proportion of the faster decay implies reduced structural defects that constitute trap states, which is in line with the enlarged crystallite sizes. The reduced defect density with NHE was also confirmed from space charge limited current measurements of the films (Supplementary Fig. 16), where the trap filling voltage ($V_{TFL}$) gradually decreased from 0.618 to 0.445 to 0.415 to 0.332 V for the control, 1P, 3P, and 3F, respectively. Correspondingly, the estimated defect density of the films was significantly decreased from $1.08 \times 10^{16}$ cm$^{-3}$ for the control film to $5.80 \times 10^{15}$ cm$^{-3}$ for the 3F film. Moreover, the significantly elongated $\tau_2$s from 32.0 ns of the control film to 1212.6 ns for 1P, 1509.1 ns for 3P, and 1862.1 ns for 3F is indicative of much longer charge carrier lifetimes in the bulk perovskite[30]. From the absorption and ultraviolet (UV) photoelectron spectroscopy (UPS) measurements, we found that FPEA$_2$PbI$_4$ forms a more

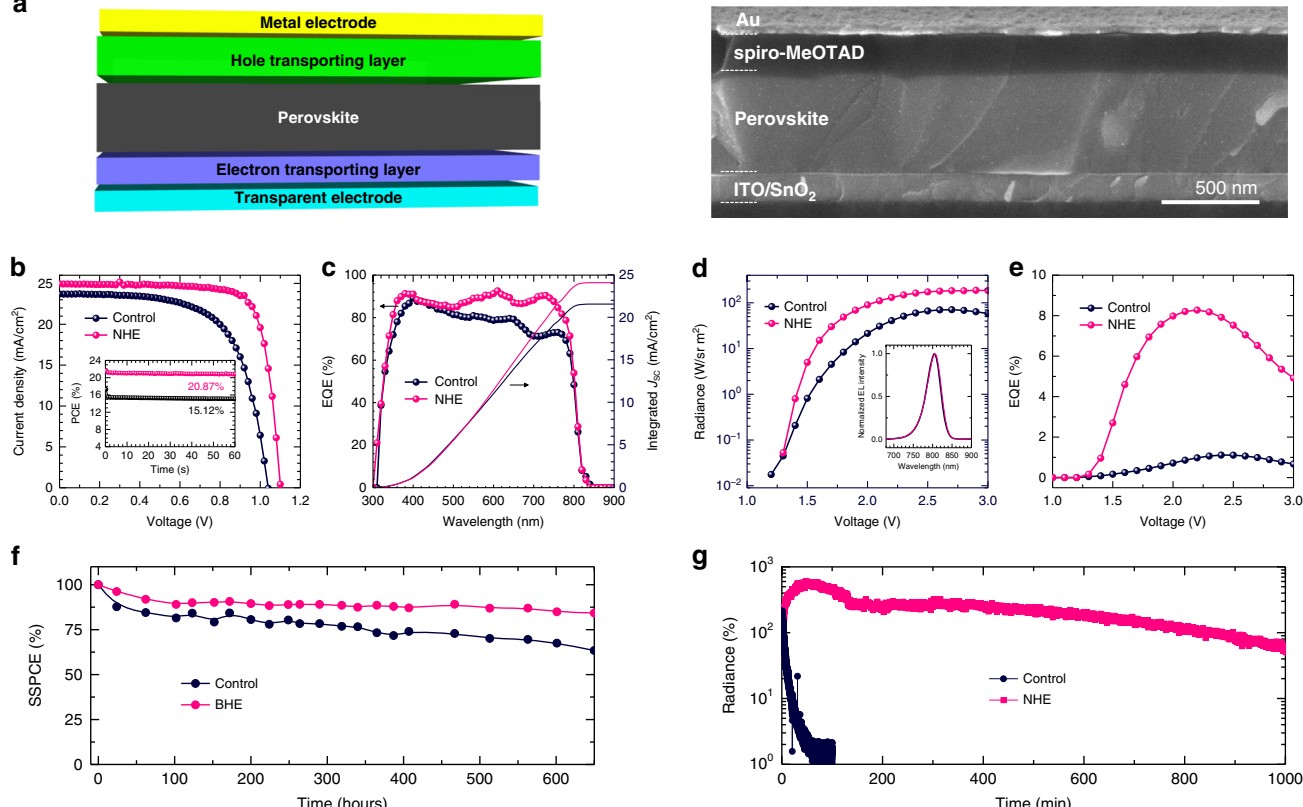

**Fig. 4 Proof-of-concept devices. a** Device structure schematics and corresponding cross-sectional scanning electron microscopy (SEM) image of the solar cell device based on the FAPbI$_3$ film with nano heteroepitaxy (NHE, with 3.33 mol% FPEA$_2$PbI$_4$). **b** Current density–voltage (J–V) and **c** external quantum efficiency (EQE) curves of solar cell devices based on a bare FAPbI$_3$ film (control) and a FAPbI$_3$ film with NHE. Inset of **b** shows the steady-state power conversion efficiencies (SSPCEs) measured at maximum power points. **d** Voltage–radiance curves of the light emitting diode (LED) devices, and **e** corresponding EQE curves of the LED devices based on the control and NHE films. Inset of **d** shows the electroluminescence spectra of the LED devices. Operational stability measurements of the **f** solar cell (normalized SSPCE) and **g** LED devices (normalized radiance) based on the control and NHE films.

ideal type I band alignment with larger energy offsets of the conduction band minimum and valence band maximum with respect to that of α-FAPbI$_3$, in comparison to the case with PEA$_2$PbI$_4$ (Supplementary Figs. 17 and 18). This will more effectively repulse the carriers from the defective grain boundaries to elongate the carrier lifetime[15]. Consequently, the average PL lifetime (τ$_{avg}$) of the control film was significantly elongated from 20.6 to 750.8 ns for 1P, 1090.7 ns for 3P, and 1613.6 ns for 3F.

**Proof-of-concept devices**. This NHE approach to grow a high-quality α-FAPbI$_3$ films was not limited by substrate choice. Indeed, it can be replicated on different substrates such as glass, ITO, polyethylene naphthalate, SnO$_2$, fluorine-doped tin oxide, and silicon (Supplementary Fig. 19), which demonstrates its universality and applicability to be used for different device applications. We fabricated proof-of-concept solar cell and LED devices to show the versatility of the high-quality NHE–FAPbI$_3$ films. The thickness of the NHE–FAPbI$_3$ film is readily controlled in the range of few tenths of nanometers as demonstrated in Supplementary Fig. 20, facilitating optimization of device performance. It should be noted that our devices are altogether MA-free (including no MACl for crystallization control), Cs-free, Rb-free, and Br-free in the perovskite composition, which is beneficial to retain the inherently low bandgap of FAPbI$_3$ and operational stability of the devices[31–33]. Figure 4a shows schematics of the device structure and cross-sectional scanning electron microscopy images of the solar cell device based on the NHE

film (3F film). The large and single-crystalline grains of the perovskite film perpendicular to the substrate is visible from the cross-sectional image. Current density–voltage (J–V) curves of the solar cell devices are shown in Fig. 4b, and the photovoltaic parameters are summarized in Table 1. The device based on the control film showed a short-circuit current density (J$_{SC}$) of 23.69 mA/cm$^2$, open-circuit voltage (V$_{OC}$) of 1.040 V, and fill factor (FF) of 0.649, corresponding to a power conversion efficiency (PCE) of 15.99%. Based on the NHE film, the PCE was significantly improved to 21.64% (35.3% improvement) with a J$_{SC}$ of 24.93 mA/cm$^2$, V$_{OC}$ of 1.101 V, and FF of 0.788. The measured PCE is competitive relative to previously reported devices based on MA-free, or MA- and Br-free perovskite compositions (Supplementary Table 5). The improved J$_{SC}$ was correlated with the improved external quantum efficiency (EQE) of the device, where the integrated J$_{SC}$s were calculated to be 21.55 and 24.07 mA/cm$^2$ for the control and NHE devices, respectively. The steady-state PCEs of the control and NHE devices were measured to be 15.12% and 20.87%, respectively (inset of Fig. 4b). The higher shunt resistance and lower series resistance of the device with 3F film calculated from the dark current measurement were correlated with the improvement in FF, whereas the lower ideality factor and saturation current were indicative of reduced non-radiative charge carrier recombination (Supplementary Fig. 21)[34], which contributed to the improved V$_{OC}$[35]. The LED with the NHE film as the emitting layer (EML) also demonstrated largely improved electroluminescent characteristics compared to the control device (Fig. 4d, e). The normalized electroluminescence

**Table 1 Measured performance parameters of perovskite solar cells and light-emitting diodes based on the control and NHE films.**

| | Solar cells | | | | | Light-emitting diodes | | |
|---|---|---|---|---|---|---|---|---|
| | $J_{SC}$ (mA/cm$^2$) | $V_{OC}$ (V) | FF | PCE (%) | T80 lifetime (h) | Maximum Radiance (W/sr m$^2$) | Maximum EQE (%) | T50 lifetime (min) |
| Control | 23.69 | 1.040 | 0.649 | 15.99 (15.12) | 726.3 | 70.78 | 1.11 | 5.7 |
| NHE | 24.93 | 1.101 | 0.788 | 21.64 (20.87) | 2025.6 | 187.7 | 8.28 | 1042.4 |

Short-circuit current density ($J_{SC}$), open-circuit voltage ($V_{OC}$), fill factor (FF), and power conversion efficiency (PCE) of the solar cells. Maximum radiance and external quantum efficiency (EQE) of the light-emitting diodes. The PCE values in parenthesis are steady-state PCEs measured at maximum power points. The T80 and T50 lifetimes of the solar cells and light-emitting diodes were extracted from Fig. 4f, g, respectively.

spectra of the devices based on the control and NHE films were almost identical (inset of Fig. 4d), whereas the NHE device showed much greater radiance over the whole operating voltage range. The maximum radiance of the device substantially increased from ~70.78 W sr$^{-1}$ m$^{-2}$ for the control device to ~187.7 W sr$^{-1}$ m$^{-2}$ by using the NHE EML (165.2% improvement). The lower current density of the 3F device than that of the control is due to more effective charge blocking within the NHE EML, enabled by the grain boundary layered perovskite formation. (Supplementary Note 4 and Supplementary Fig. 22). Consequently, the NHE LED device had a largely improved EQE, reaching ~8.28% at a low operating voltage (2.2 V), which is more than seven-fold enhanced compared to that of the control device (~1.11% at 2.4 V). The considerable performance enhancements for both the solar cell and LED devices can be mainly attributed to enhanced crystallinity and thus lower defect density of the perovskite films, resulting in the suppressed non-radiative recombination of the charge carriers. This might promote effective charge carrier collection and enhanced radiative recombination in the solar cells and LEDs, respectively. Also, the grain boundary layered perovskites forming a type-I band alignment would additionally help to repulse the photo-generated or injected charge carriers from the defective grain boundaries, which will be helpful for the performance of both solar cells and LEDs.

The operational stabilities of the solar cells and LED devices were evaluated in Fig. 4f, g, respectively. The stability of the solar cells was tested under 1 sun illumination (no UV filter) at open-circuit condition, where a more accelerated degradation is expected compared to the case of biasing the device at its actual maximum power point due to the stronger built-in potential and increased charge carrier recombination in the device[36]. Rapid initial decays in the SSPCEs of 20.3% and 8.6% were observed for the control and NHE devices, respectively. After the initial decay, a slower decay with an almost linear profile followed. After 650 h of continuous illumination, the SSPCE of the control device degraded to 63.4% of its initial value, while the NHE device retained 84.2% of its initial SSPCE. The T80 lifetimes (time at which the SSPCE degrades to 80% of its initial value) of the devices were then estimated with the assumption that the linear degradation profile is maintained[37]. The T80 lifetime for the control device was estimated to be 733.4 h, and was significantly enhanced by 178% to 2040.4 h for the device based on the NHE film. To evaluate the operational stability of the LEDs, the radiance evolutions of the devices were recorded under a constant-voltage operation (constant voltage that initially emits radiance ~10 W sr$^{-1}$ m$^{-2}$). The LED device based on the NHE film also showed a highly elongated lifetime as shown in Fig. 4g. The T50 lifetime (time at which the radiance degrades to 50% of its initial value) of the devices was significantly enhanced from 5.7 min for the control device to 1042.4 min for the NHE device. Noting that the active migration of ionic defects in the perovskite is mainly responsible for degradation of the devices under

operational conditions[38–41], we speculate that the higher crystallinity with lower defect density of the NHE films is the origin of the improved operational stabilities of the solar cells and LED devices. From temperature-dependent conductivity measurements in Supplementary Fig. 23, the activation energy for ion migration in the NHE films was determined to be 1.435 eV, which is much higher than that measured for the control films (0.516 eV), correlating with the retarded degradation dynamics. The much higher activation energy for ion migration in the NHE film can be attributed to significantly its enhanced crystallinity and reduced defect density.

## Discussion

In this work, we demonstrated kinetic-controlled and substrate-tolerant local epitaxial growth of FAPbI$_3$ perovskite crystals. The layered perovskite-templated epitaxial crystal growth was induced during the solid-state phase transformation of hexagonal FAPbI$_3$ into its cubic perovskite polymorph. The phase conversion kinetics was retarded by the induced strain at the hetero-interface between the layered perovskites and FAPbI$_3$. Our first-principles calculations revealed that the hexagonal-to-cubic conversion energy barrier was controlled by a synergistic effect between the induced strain and entropy. The kinetic-controlled NHE facilitated the growth of 10-fold enlarged FAPbI$_3$ perovskite crystals with a reduced defect density and strong preferred orientation. Resultingly, our proof-of-concept solar cell and LED devices showed efficiencies and stabilities superior to the controls. We believe our approach will provide new insights to innovatively reduce the defect density in thin films of perovskites and other semiconducting materials using a simple, cheap and versatile method to further improve their stability and performance.

## Methods

**Perovskite precursor solution**. A precursor solution for bare FAPbI$_3$ (control) film was prepared by dissolving equimolar amount of HC(NH$_2$)$_2$I (FAI, Dyesol), PbI$_2$ (TCI, 99.99%), and N-methyl-2-pyrrolidone (NMP, Sigma-Aldrich, anhydrous, 99.5%) in N,N-Dimethylformamide (DMF, Sigma-Aldrich, anhydrous, 99.8%). In a typical process, 172 mg of FAI, 461 mg of PbI$_2$ and 99 mg of NMP were dissolved in 560 mg of DMF. For the films with layered perovskite, the corresponding FAPbI$_3$ precursors were replaced with the same molar amount of the precursors for the layered perovskite. Typically, the precursor solution for 1P film was prepared by dissolving 8.2 mg of phenethylammonium iodide (PEAI, Sigma-Aldrich, 98%), 166.4 mg of FAI, 453.4 mg of PbI$_2$, and 97.4 mg of NMP in 560 mg of DMF. The precursor solution for 3P film was prepared by dissolving 16.4 mg of phenethylammonium iodide (PEAI, Sigma-Aldrich, 98%), 160.9 mg of FAI, 446.1 mg of PbI$_2$ and 95.8 mg of NMP in 560 mg of DMF. The precursor solution for 3F film was prepared by dissolving 17.2 mg of 4-fluoro phenethy-lammonium iodide (FPEAI, Greatcellsolar), 160.9 mg of FAI, 446.1 mg of PbI$_2$, and 95.8 mg of NMP in 560 mg of DMF.

**Fabrication of perovskite films**. All the films were prepared inside a glove box filled with dry air (Dew point around −39 °C). The perovskite precursor solution was filtered with 0.2 μm pore sized PTFE syringe filter before use. The precursor solution was spin-coated at 4000 rpm for 20 s to which 0.15 mL of diethyl ether (anhydrous, >99.0%, contains BHT as stabilizer, Sigma-Adrich) was dropped after 10 s. The resulting film was heat-treated at 150 °C for 20 min (the annealing time

for bare FAPbI$_3$ film was 10 min due to thermal degradation with longer annealing time).

**In situ GIWAXS measurement**. The GIWAXS data were collected at beamline 7.3.3 of the Advanced Light Source, Lawrence Berkeley National Laboratory. All the films were prepared on the silicon wafer with 100 nm silicon oxide without annealing. During the measurement, samples were heated on the heating stage at 150 °C. The diffraction patterns were collected every 4 seconds with each exposure extending for 2 s at a fixed incident angle of 0.3°. All measurements were conducted under a helium atmosphere to reduce air scattering.

**XRD measurements**. XRD patterns were measured by using X-ray diffractometer (PANalytical) with Cu kα radiation at a scan rate of 8°/min with step size of 0.0083556°. For the higher resolution XRD measurement (Fig. 3a, b), the scan rate and step size were adjusted to 2°/min and 0.0020889°, respectively. A high-resolution Jordan Valley D1 X-ray diffractometer with Cu Kα radiation and incident parallel beam optics was employed to obtain both $2\theta{:}\omega$ and pole figure measurements. Pole figures were measured by fixing $2\theta$ at the proper diffraction angle and taking $\varphi$ scans (in-plane rotation angle) while varying $\chi$ (tilt angle whose axis of rotation is orthogonal to $\omega$ and $\varphi$). The $\varphi$ scans ranged from 0° to 360° with a step size of 1° and $\chi$ ranged from 0° to 90° with a step size of 2°. The final pole figures were then plotted with the experimentally measured intensity as a function of both $\varphi$ and $\chi$.

**Microstructure analyses (SEM, TEM, and AFM)**. Cross-sectional morphology of the devices were investigated by SEM (Nova Nano 230). The cross sectional surface of the sample was coated with a ca. 1 nm-thick gold layer using a sputter to prevent the charging during the measurement. TEM images were taken by Titan Krios (FEI). The perovskite film was scratched off from the substrate and dispersed in toluene, which was dropped on a TEM grid. Accelerating voltage of 300 kV was used for the measurement. In order to perform in situ TEM measurements, a cross-sectional TEM sample was lift out from the perovskite film and welded on a Cu lift-out grid using focused ion beam (FIB, Tescan GAIA3). Before the FIB sample preparation, a 5 nm Ir layer was sputtered on the top of the film as the first protection layer. During FIB, electron beam induced Pt was deposited on the sample followed by iron beam induced Pt. During the thinning process, we use 30 kV 600 pA to first thin the lamina down to 600 nm, then we switch to 15 kV 200 pA to thin it down to around 150 nm. Finally, we cleaned the film using 3 kV 90 pA to minimize the ion beam damage. In-situ TEM measurement was carried out on JEM-ARM300F Grand ARM (JEOL) under 300 kV TEM mode, and a double tilt heating holder (Model 652, Gatan) without cooling water attachment for heating with improved stability. Atomic force microscopic (AFM) images were measured by Bruker dimension Fast Scan using a 1 ohm silicon tip (OTESPA, Bruker).

**Absorption and photoluminescence measurements**. UV–visible absorption spectra were obtained by U-4100 spectrophotometer (Hitachi) equipped with integrating sphere. The monochromatic light was incident to the sample from a substrate side. Steady-state PL signal was obtained by a Horiba Jobin Yvon system. The sample was excited by A 640 nm monochromatic laser. Time-resolved PL decay profiles were measured by using a Picoharp 300 with time-correlated single-photon counting capabilities. The films were excited by a 640 nm pulsed laser with a repetition frequency of 100 kHz provided by a picosecond laser diode head (PLD 800B, PicoQuant) triggered by a pulse generator. The energy density of the excitation laser was ca. 1.4 nJ/cm$^2$.

**Space charge limited current measurement**. The sample was prepared with architecture of ITO/perovskite film/Au. The current–voltage scan was performed under dark with scan rate of 0.005 V/s. The trap filling voltage was determined by extrapolation of ohmic regime ($I \propto V$) and trap-filled regime ($I \propto V^{n>3}$). All the curves showed a trap-free Child's regime afterward ($I \propto V^2$).

**UPS measurement**. UPS measurement was carried out using Kratos UV photo-electron spectrometer. He I (21.22 eV) source was used as an excitation source. The perovskite films were coated on an ITO substrate and grounded using a silver paste to avoid the charging during the measurement.

**DFT calculations**. The DFT calculations for charge distribution were performed using Gaussian09 program package[42]. The geometry optimizations were carried out at the B3LYP-D3 level of theory with the 6-311++G(d,p) basis set in gas phase. The vibrational frequencies were computed at the same level to confirm that the optimized structures are at an energy minimum (zero imaginary frequencies). Electrostatic potentials on isosurfaces and atomic charges of all atoms were computed using B3LYP-D3 level of theory with 6-31G(d) basis set in gas phase according to Hirshfeld[43–45] population analysis. Dipole moments were computed using the same level of theory according to Merz–Singh–Kollman[46,47] scheme. Choice of schemes was based on preceding literature[48,49]. The diagrams were generated using GaussView 5[50]. For phase conversion energetics, all the calculations are based on first-principles DFT using the Vienna ab initio simulation

package[51] with the projector augmented wave method; a generalized gradient approximation in the form of Perdew–Burke–Ernzerhof functional for the exchange-correlation functional[52,53], and an energy cutoff of 400 eV is employed for all the calculations. To consider the van der Waals interactions, the dispersion correction by using Grimme's DFT–D3 scheme is considered[54]. For the cubic FAPbI$_3$, a $4 \times 4 \times 4$ Γ-centered Monkhorst–Pack sampling mesh is used to get the accurate lattice constants. For the hexagonal FAPbI$_3$ and the layered perovskite, the k-points sampling are $3 \times 3 \times 3$ and $2 \times 4 \times 4$. For surfaces, k-point mesh is taken to be 1 for the direction of the surface. The surfaces were modeled by a periodic slab consisting of at least five atomic layers, separated by at least 20 Å of vacuum in the surface normal direction. During our calculations, all atoms are fully relaxed until the residual forces on each atom are less than 0.02 eV/Å. The surface energy is defined as:

$$\gamma = \frac{1}{2A}(E_{slab} - n\mu_{mol} - m\mu_{Pb} - l\mu_I),$$

where $E_{slab}$ is the total energy of the slab model with two surfaces, A is the surface area per unit cell, and each quantities satisfy the relations, $\mu_{mol} + \mu_{Pb} + 3\mu_I = E_{cubic}$ for cubic phase and $\mu_{mol} + \mu_{Pb} + 3\mu_I = 1/2E_{hex}$ for hexagonal phase. Among the high symmetry surfaces of cubic and hexagonal phase of FAPbI$_3$ investigated in this study, the (111) surface of cubic FAPbI$_3$ and (001) surface of hexagonal FAPbI$_3$ have the highest surface energies (see Fig. 2) with the energy close to ~6 eV/nm$^2$.

**Fabrication of perovskite solar cells**. ITO glass was cleaned by successive sonication in detergent, deionized (DI) water, acetone and 2-propanol for 15 min respectively. The cleaned substrates were further treated with UV-ozone to remove the organic residual and enhance the wettability. A 30 mM SnCl$_2 \cdot 2H_2O$ (Aldrich, >99.995%) solution was prepared in anhydrous ethanol (Decon Laboratories Inc.) and filtered by 0.2 μm syringe filter before use. The solution was spin-coated on the cleaned ITO substrate at 3000 rpm for 30 s, which was followed by heat-treatment at 150 °C for 30 min. After cooling down to room temperature, the spin-coating process was repeated one more time, and the resulting film was annealed at 150 °C for 5 min and 180 °C for 1 h. The SnO$_2$ coated ITO glass was further treated with UV-ozone before fabrication of the perovskite layer. Perovskite films were prepared according to the procedure described in an above section. The spiro-MeOTAD solution was prepared by dissolving 85.8 mg of spiro-MeOTAD (Lumtech) in 1 mL of anhydrous chlorobenzene (99.8%, Sigma-Aldrich) to which 33.8 μl of 4-tert-butylpyridine (96%, Aldrich) and 19.3 μl of Li-TFSI (99.95%, Aldrich, 520 mg/mL in acetonitrile) solution were added. The spiro-MeOTAD solution was spin-coated on top of the perovskite layer at 3000 rpm for 20 s by dropping 17 μl of the solution on the spinning substrate. The device was completed by thermal evaporation of ca. 100 nm-thick silver or gold layer at 0.5 Å/s to be used as an electrode. The size of active layer was 0.130 cm$^2$.

**Fabrication of perovskite LED**. ITO substrate was cleaned and treated with UV-ozone with same procedure for fabrication of solar cell devices. The 1.9 wt% colloidal ZnO solution was prepared by diluting the stock ZnO nanoparticle solution (Aldrich, 2.5 wt% in isopropanol and propylene glycol) with anhydrous iso-propanol (Sigma-Aldrich, 99.5%). The ZnO solution was spin-coated on the ITO substrate at 5000 rpm for 30 s, which was followed by drying at 100 °C for 10 min. The ZnO coated ITO substrate was further treated with UV-ozone for 10 min before spin-coating of perovskite solution. The DMF amount for the perovskite solution was adjusted to 660 mg. The perovskite solution was spin-coated at 4000 rpm for 25 s, to which 0.2 mL of anhydrous chloroform (Sigma-Aldrich, >99.0%) was dropped after 10 s of spinning. The resulting film was annealed at 80 °C for 1 min followed by 150 °C for 1 min. On top of the perovskite layer, 0.5 wt.% of Poly [N,N′-bis(4-butylphenyl)-N,N′-bis(phenyl)-benzidine] (Poly-TPD, Lumtec) solution in chlorobenzene was spin-coated at 4000 rpm for 30 s. The device was completed with thermal evaporation of 10 nm-thick molybdenum trioxide and 100 nm-thick silver layers. The size of active layer was 0.100 cm$^2$.

**Photovoltaic device characterization of solar cells**. Current density–voltage ($J$–$V$) curves of the solar cell devices were measured using Keithley 2401 source meter under simulated one sun illumination (AM 1.5 G, 100 mW/cm$^2$) in ambient atmosphere. The one sun illumination was generated from Oriel Sol3A with class AAA solar simulator (Newport), in which light intensity was calibrated by NREL-certified Si photodiode equipped with KG-5 filter. Typically, the $J$–$V$ curves were recorded at 0.1 V/s (between 1.2 and −0.1 V with 65 data points and 0.2 s of delay time per point). During the measurement, the device was covered with metal aperture (0.100 cm$^2$) to define the active area (the whole active area was around 0.130 cm$^2$). The Dark $J$–$V$ curve was obtained with same measurement setup with slower scan rate (0.01 V/s). All the devices were measured without preconditioning such as light-soaking and applied bias voltage. Steady-state power conversion efficiency was calculated by measuring stabilized photocurrent density under constant bias voltage. The external quantum efficiency was measured using specially designed system (Enli tech) under AC mode (frequency = 133 Hz) without bias light.

**Characterization of LEDs**. A Keithley 2400 source meter and silicon photodiode (Hamamatsu S1133-14, Japan) calibrated by a PR650 spectroradiometer (Photo Research) were used to measure Current–voltage–radiance characteristics of perovskite LEDs. Electroluminescence spectra were recorded by Horiba Jobin Yvon system, and used to calculate radiance and external quantum efficiency. All the devices were assumed as Lambertian emitter in the calculation. For measurement of operational stability, the Keithley 2400 source meter and photodiode (Hamamatsu S1133-14) was used to apply constant bias to record photocurrent response, respectively. Capacitance versus voltage measurement was carried out by using an Agilent 4284A LCR meter controlled by a LabView program. DC bias was swept from 0 to 2 V with 20 mV AC oscillation amplitude and 1 kHz frequency.

**Operational stability measurement of the devices**. The solar cell and LED devices were encapsulated under nitrogen atmosphere by using glass slit and UV curable sealant. All the devices were kept under the nitrogen atmosphere for at least 12 h before and after encapsulation. The encapsulated solar cell devices were exposed to continuous 0.9 sun ($90 \pm 5$ mW/cm$^2$) illumination under open circuit condition, and the steady-state power conversion efficiency of the devices was periodically measured under 1 sun illumination with different exposure time. The encapsulated LEDs were kept operating under constant applied bias with source meter (Keithley 2400), and emitted light intensity of LEDs was recorded every 30 s with photodiode (Hamamatsu S1133-14) calibrated by PR650 (Photo Research) spectro-radio meter.

**Temperature-dependent conductivity measurement**. The temperature-dependent conductivity measurement was carried out using a commercial probe station (Lakeshore, TTP4) in which temperature of the device was controlled by thermoelectric plate and flow of liquid nitrogen. The electrical measurement was conducted with a source/measurement unit (Agilent, B2902A). A lateral device with architecture of Au/perovskite (100 μm)/Au was used for the measurement.

**Reporting summary**. Further information on research design is available in the Nature Research Reporting Summary linked to this article.

## Data availability
The authors declare that the data supporting the findings of this study are available within the paper and its Supplementary information files.

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

## Acknowledgements

This work was supported by the U.S. Department of Energy's Office of Energy Efficiency and Renewable Energy (EERE) under the Solar Energy Technologies Office under award number DE-EE0008751. A portion of research was supported by the Center for Nanophase Materials Sciences (first-principles modeling), which is a DOE Office of Science User Facility; by the Creative Materials Discovery Program through the National Research Foundation of Korea (NRF) (phase diagram) funded by the Ministry of Science, ICT and Future Planning (NRF-2016M3DIA1919181). This research used resources of the Oak Ridge Leadership Computing Facility and the National Energy Research Scientific Computing Center, DOE Office of Science User Facilities. This work was also supported by the National Research Foundation of Korea (NRF) grant funded by the Korea government (MIST) under contract Nos. 2020R1F1A1067223 and 2020R1C1C1008485.

## Author contributions

J.-W.L. conceived an idea and led overall project under supervision of Y.Y. J.-W.L. and S.T. performed most of the experiment and analyze the data. T.-H.H. assisted in the stability measurement of LED devices. R.W. and C.Z. helped in GIWAXS measurements. L.Z., C.P., M.Y., and S.N. conducted the computational works. C.C., M.X., Y.H., and X.P. carried out the TEM characterization. M.E.L. and M.S.G. performed the pole figure measurement. S.L. investigated temperature-dependent conductivity of the films. All the authors discussed and commented on the paper.

## Competing interests

The authors declare no competing interests.
