## [Peer Review File · Nature Communications]

Reviewers' Comments:

Reviewer #1:

Remarks to the Author:

I have carefully read the manuscript, the previous reviewers' comments, and the responses and changes from the authors. Overall, I think this is good study. The impact of using PEAI and FPEAI based additives on FAPbI3 device performance and stability is quite significant and impressive. The authors have also done a good job to clarify the underlying mechanism in response to previous reviewers' comments. This manuscript is publishable after the authors address a few comments below.

1. For the stability discussion, T80 of 2040.4 h was extrapolated based on about 700 h measurement. This assumes that the degradation process is not changed for 2040.4 h. This may or may not be true. Thus, I do not support extrapolation of the stability data. Please report based on the actual measurement.
2. The authors emphasized that the devices in this study were MA-free, Cs-free, Rb-free, and Br-free. What is the advantage for targeting these characteristics for developing perovskites? Please clarify.
3. Related to 2 above, ref. 14 in the response file showed 24.7% for pure FAPbI3 devices. MDACI2 was used as the additive there. In this study, PEAI and FPEAI based low dimensional structures were used. So, ref. 14 is quite similar to the current study. FAPbI3 is the targeted material for both studies. Thus, Table R1 and the discussion in the response file need to be modified.

Reviewer #2:

Remarks to the Author:

This work by J.-W. Lee et al reports a high performance FAPbI3 solar cells with epitaxial nanoscale layered perovskite FPEA2PbI4-FAPbI3 heterointerfaces. It is proposed that a small amount of 2D layered perovskites exists in the mixture and this 2D perovskite nanocrystals serve as nucleation sites for the subsequent epitaxial growth of FAPbI3 domain. This unique growth mechanism leads to improved crystal quality and solar cell device performance. Very careful structural and device characterizations have been carried out. It looks like the manuscript has been reviewed previously by two referees and constructive comments have been provided. In this revised version, the authors have added substantial new data to clarify their claims and enhanced the novelty of the work. This work is original and of high intellectual merit. I recommend its publication in nature communications after addressing two minor comments: 1. It is impressive to have the high quality in-situ TEM cross-sectional image. This is not easy, as halide perovskite materials are very sensitive to heat and electron beam irradiation. From these images (e.g., Fig S5), the distance between the lattice fringes are 7.25 Å, which is attributed to (002) planes of the 2D perovskite. However, the layer distance for the n=1 2D perovskite is ~15.2 Å (Fig S7f), which is more than double of the "7.25 Å" spacing. It will be great if the authors could comment / clarify what is the 7.25 Å corresponding to (e.g., distance between two layers of Pb atom in the 2D structure?). 2. Epitaxial growth of halide perovskites (Nature 2020, 577, 209), especially lateral epitaxial in 2D perovskites (Nature 2020, 580, 614) have been recently demonstrated, these studies could be mentioned briefly in the introduction to set a broader context. 3.

The black (yellow) FAPbI₃ phase is some times called "alpha(delta)-FAPbI₃", "cubic(hexagonal) alpha(delta)-FAPbI₃", and "cubic(hexagonal)-FAPbI₃" across the main text and SI (including the figure captions). It is better to be consistent with the naming to avoid confusion.

Response to the reviewers' comments

We appreciate the reviewers' valuable comments and suggestions. We have revised our manuscript to reflect them, and have prepared point-by-point responses to their questions. The revised parts in the manuscript are marked in blue.

Reviewer #1 (Remarks to the Author):

I have carefully read the manuscript, the previous reviewers' comments, and the responses and changes from the authors. Overall, I think this is good study. The impact of using PEAI and FPEAI based additives on FAPbI₃ device performance and stability is quite significant and impressive. The authors have also done a good job to clarify the underlying mechanism in response to previous reviewers' comments. This manuscript is publishable after the authors address a few comments below.

1. For the stability discussion, T80 of 2040.4 h was extrapolated based on about 700 h measurement. This assumes that the degradation process is not changed for 2040.4 h. This may or may not be true. Thus, I do not support extrapolation of the stability data. Please report based on the actual measurement.

Ans) We thank the reviewer for the comment. To address the reviewer's concern, we have modified the stability discussion to report both the actual measurement result and the T80 data, and the assumptions made to estimate the T80 was clarified.

Page 13, line 17

“Rapid initial decays in the SSPCEs of 20.3% and 8.6% were observed for the control and NHE devices, respectively. After the initial decay, a slower decay with an almost linear profile followed. After 650 h of continuous illumination, the SSPCE of the control device degraded to 63.4% of its initial value, while the NHE device retained 84.2% of its initial SSPCE. The T80 lifetimes (time at which the SSPCE degrades to 80% of its initial value) of the devices were then estimated with the assumption that the linear degradation profile is maintained.¹”

2. The authors emphasized that the devices in this study were MA-free, Cs-free, Rb-free, and Br-free. What is the advantage for targeting these characteristics for developing perovskites? Please clarify.

Ans) We thank the reviewer for the helpful comment. Pure FAPbI₃ has the lowest bandgap among Pb-based perovskites (1.48 eV), which is close to the ideal bandgap for solar cells dictated by the Shockley-Queisser limit (~1.4 eV). However, the difficulty in fabricating a phase-pure and highly crystalline α -phase FAPbI₃ film, as well as the thermodynamic instability of the desired α -phase, hinders the realization of high efficiency perovskite solar cells based on pure FAPbI₃. Incorporation of MA, Cs, Rb, and Br have been reported to stabilize the desired α -FAPbI₃ phase,²⁻⁵ but at a cost of increasing the bandgap to sacrifice the device photocurrent.

Furthermore, incorporation of MA cation was found to be detrimental to the operational stability of the perovskite solar cells because of its thermal instability.^{2,6} In this regard, our study demonstrated a unique approach to fabricate high-quality and phase-pure α -FAPbI₃ with enhanced stability without changing its inherent bandgap. We believe that further optimization of our approach will provide an opportunity to realize perovskite solar cell devices with even higher efficiency and better longevity. To reflect the reviewer's comment, we have added the following sentences in the revised manuscript.

In page 12, line 12

“It should be noted that our devices are altogether MA-free (including no MAI for crystallization control), Cs-free, Rb-free, and Br-free in the perovskite composition, which is beneficial to retain the inherently low bandgap of FAPbI₃ and operational stability of the devices.^{2,5,6}”

3. Related to 2 above, ref. 14 in the response file showed 24.7% for pure FAPbI₃ devices. MDACl₂ was used as the additive there. In this study, PEAI and FPEAI based low dimensional structures were used. So, ref. 14 is quite similar to the current study. FAPbI₃ is the targeted material for both studies. Thus, Table R1 and the discussion in the response file need to be modified.

Ans) We thank the reviewer for the comment. In the revised manuscript, we did not distinguish the work reporting the MDACl₂ based FAPbI₃ with our study. We only specified that it contained Cl in their composition.

In Page 12, line 24

“The measured PCE is competitive relative to previously reported devices based on MA-free, or MA- and Br-free perovskite compositions (**Supplementary Table 5**).”

Table S5 | Summary of reported high-performance MA-free, or MA- and Br-free perovskite solar cells.

Composition	Device structure	E_g (eV)	V_{OC} (V)	J_{sc} (mA cm ⁻²)	FF	PCE (%)	Ref
This work: NHE-FAPbI ₃	ITO/SnO ₂ /perovskite/spiro- OMeTAD/Au	1.48	1.10	24.93	0.79	21.6	-
Literature references for MA-free solar cells:							

$(\text{Cs}_{0.15}\text{FA}_{0.85})\text{Pb}(\text{I}_{0.9}\text{Br}_{0.1})_3$	ITO/PTAA/PFN-P2/perovskite/ LiF/C60/BCP/Cu	1.58	1.11	23.19	0.80	20.7	7
$\text{Cs}_{0.15}\text{FA}_{0.85}\text{Pb}(\text{I}_{0.9}\text{Br}_{0.1})_3$	FTO/bl-TiO ₂ /perovskite/PbS/ spiro-OMeTAD/Au	1.58	1.15	23.06	0.80	21.1	8
$\text{BA}_{0.05}(\text{Cs}_{0.17}\text{FA}_{0.83})_{0.95}\text{Pb}(\text{I}_{0.8}\text{Br}_{0.2})_3$	FTO/SnO ₂ /PCBM/perovskite/ Spiro-OMeTAD/Ag	1.61	1.14	22.70	0.80	20.6	9
$(\text{Cs}_{0.17}\text{FA}_{0.83})\text{Pb}(\text{I}_{0.89}\text{Br}_{0.08}\text{Cl}_{0.03})_3$	FTO/bl-TiO ₂ /mp-TiO ₂ /SnO ₂ / perovskite/spiro-OMeTAD/Au	1.58	1.12	23.28	0.78	20.5	10
$(\text{CsPbBr}_3)_{0.06}(\text{FAPbI}_3)_{0.94}$	FTO/bl-TiO ₂ /mp-TiO ₂ /perovskite/spiro-OMeTAD/Au	1.55	1.15	24.52	0.78	21.8	11
$\text{Cs}_{0.17}\text{FA}_{0.83}\text{Pb}(\text{I}_{0.8}\text{Br}_{0.2})_3$	ITO/PTAA/PFN-Br/perovskite/ C ₆₀ /BCP/Cu	1.61	1.15	22.58	0.81	21.1	12
Literature references for MA-free, Br-free solar cells:							
$\text{Rb}_{0.05}\text{FA}_{0.95}\text{PbI}_3$	FTO/bl-TiO ₂ /mp-TiO ₂ /perovskite/spiro-OMeTAD/Au	1.53	1.07	23.93	0.67	17.2	13
$\text{Cs}_{0.1}\text{FA}_{0.9}\text{PbI}_3$	FTO/bl-TiO ₂ / /perovskite/spiro-OMeTAD/Au	1.48	1.07	23.40	0.76	19.0	2
$\text{Rb}_{0.05}\text{Cs}_{0.10}\text{FA}_{0.85}\text{PbI}_3$	FTO/SnO ₂ /PCBM-PMMA/ perovskite/Spiro-OMeTAD/Au	1.53	1.08	25.06	0.76	20.4	6
$\text{Cs}_{0.05}\text{FA}_{0.95}\text{PbI}_3$	ITO/PTAA/PFN-P2/perovskite/ LiF/C60/BCP/Cu	1.50	1.05	25.10	0.75	19.8	14
$\text{Cs}_{0.02}\text{FA}_{0.98}\text{PbI}_3$	ITO/SnO ₂ /perovskite/spiro-OMeTAD/Au	1.48	1.10	23.98	0.77	20.2	15

$[\text{PEA}_2\text{PbI}_4]_{0.167}[\text{Cs}_{0.02}\text{FA}_{0.98}\text{PbI}_3]_{0.9833}$	ITO/SnO ₂ /perovskite/spiro-OMeTAD/Au	1.48	1.126	24.44	0.740	21.06	16
Literature references for MA-free, Br-free solar cells, with Cl (e.g. MACl)							
FAPbI ₃ :(MDACl ₂) _x (x = 3.8 mol%)	FTO/bl-TiO ₂ /mp-TiO ₂ /perovskite/passivation layer/spiro-OMeTAD/Au	1.47	1.14	26.50	0.82	24.7	17
Cs _{0.2} FA _{0.8} PbI ₃ -(Cl)	FTO/bl-TiO ₂ /mp-TiO ₂ /perovskite/spiro-OMeTAD/Au	1.56	1.10	24.10	0.78	20.6	18
FAPbI ₃	FTO/SnO ₂ /Perovskite/Spiro-MeOTAD/Au	1.48	1.04	24.8	0.746	19.3	19

Reviewer #2 (Remarks to the Author):

This work by J.-W. Lee et al reports a high performance FAPbI₃ solar cells with epitaxial nanoscale layered perovskite FPEA₂PbI₄-FAPbI₃ heterointerfaces. It is proposed that a small amount of 2D layered perovskites exists in the mixture and this 2D perovskite nanocrystals serve as nucleation sites for the subsequent epitaxial growth of FAPbI₃ domain. This unique growth mechanism leads to improved crystal quality and solar cell device performance. Very careful structural and device characterizations have been carried out. It looks like the manuscript has been reviewed previously by two referees and constructive comments have been provided. In this revised version, the authors have added substantial new data to clarify their claims and enhanced the novelty of the work. This work is original and of high intellectual merit. I recommend its publication in nature communications after addressing two minor comments:

1. It is impressive to have the high quality in-situ TEM cross-sectional image. This is not easy, as halide perovskite materials are very sensitive to heat and electron beam irradiation. From these images (e.g., Fig S5), the distance between the lattice fringes are 7.25 Å, which is attributed to (002) planes of the 2D perovskite. However, the layer distance for the n=1 2D perovskite is ~15.2 Å (Fig S7f), which is more than double of the "7.25 Å" spacing. It will be great if the authors could comment / clarify what is the 7.25 Å corresponding to (e.g., distance between two layers of Pb atom in the 2D structure?).

Ans) We thank the reviewer for the constructive comment. In truth, we are not entirely sure why there is a discrepancy between the interplanar distances observed for the bulk 2D perovskite

versus 2D perovskite in the FAPbI₃ film. In fact, similar observations have been reported in previously published literatures reporting 2D/3D mixed perovskite compositions.^{16,20-22} We speculate that this is due to the presence of 2D perovskite polymorphs with different orientations of the bulky organic cation in the lattice (e.g. PEA⁺ or FPEA⁺). These organic cations are mechanically soft and form relatively weak hydrogen bonding with the PbI₆ lattice. As a result, the cations can easily change their orientation in the lattice (this situation is more probable when strain is present, induced by a lattice mismatch between the hetero phases). In fact, Chen et al. observed the co-existence of 2D perovskite polymorphs whose lattice spacing is substantially different (10.9 Å for one and 9.2 Å for the other).²² Therefore, we think that the 2D perovskite with A cations of different orientation might be formed in the hetero-phased film. We plan to do further works to clarify this in our future studies.

2. Epitaxial growth of halide perovskites (Nature 2020, 577, 209), especially lateral epitaxial in 2D perovskites (Nature 2020, 580, 614) have been recently demonstrated, these studies could be mentioned briefly in the introduction to set a broader context.

Ans) We thank the reviewer for the comment. The reference [Nature 2020, 577, 209] was already cited in the original manuscript, and we added the other [Nature 2020, 580, 614] regarding the lateral epitaxy of 2D perovskite as reference 11 in the introduction part of the revised manuscript.

3. The black (yellow) FAPbI₃ phase is some times called "alpha(delta)-FAPbI₃", "cubic(hexagonal) alpha(delta)-FAPbI₃", and "cubic(hexagonal)-FAPbI₃" across the main text and SI (including the figure captions). It is better to be consistent with the naming to avoid confusion.

Ans) We thank the reviewer for the helpful suggestion. We have modified the revised manuscript to be consistent by using the terms; “α-FAPbI₃” and “δ-FAPbI₃”

References

- 1 Tan, S. *et al.* Steric Impediment of Ion Migration Contributes to Improved Operational Stability of Perovskite Solar Cells. *Adv. Mater.* **32**, 1906995 (2020).
- 2 Lee, J. W. *et al.* Formamidinium and cesium hybridization for photo□ and moisture□ stable perovskite solar cell. *Adv. Energy Mater.* **5**, 1501310 (2015).
- 3 Park, Y. H. *et al.* Inorganic Rubidium Cation as an Enhancer for Photovoltaic Performance and Moisture Stability of HC(NH₂)₂PbI₃ Perovskite Solar Cells. *Adv. Funct. Mater.* **27**, 1605988, doi:10.1002/adfm.201605988 (2017).
- 4 Jeon, N. J. *et al.* Compositional engineering of perovskite materials for high-performance solar cells. *Nature* **517**, 476-480 (2015).

- 5 Syzgantseva, O. A., Saliba, M., Grätzel, M. & Rothlisberger, U. Stabilization of the perovskite phase of formamidinium lead triiodide by methylammonium, Cs, and/or Rb doping. *J. Phys. Chem. Lett.* **8**, 1191-1196 (2017).
- 6 Turren-Cruz, S.-H., Hagfeldt, A. & Saliba, M. Methylammonium-free, high-performance, and stable perovskite solar cells on a planar architecture. *Science* **362**, 449-453 (2018).
- 7 Chen, Y. *et al.* Thermally stable methylammonium-free inverted perovskite solar cells with Zn²⁺ doped CuGaO₂ as efficient mesoporous hole-transporting layer. *Nano Energy* **61**, 148-157 (2019).
- 8 Chen, Y. *et al.* Interfacial Contact Passivation for Efficient and Stable Cesium-Formamidinium Double-Cation Lead Halide Perovskite Solar Cells. *IScience* **23**, 100762 (2020).
- 9 Wang, Z. *et al.* Efficient ambient-air-stable solar cells with 2D–3D heterostructured butylammonium-caesium-formamidinium lead halide perovskites. *Nat. Energy* **2**, 17135 (2017).
- 10 Gao, X. X. *et al.* Stable and High Efficiency Methylammonium-Free Perovskite Solar Cells. *Adv. Mater.* **32**, 1905502 (2020).
- 11 Xie, L. *et al.* Efficient and Stable Low-Bandgap Perovskite Solar Cells Enabled by a CsPbBr₃-Cluster Assisted Bottom-up Crystallization Approach. *J. Am. Chem. Soc.* **141**, 20537-20546 (2019).
- 12 Li, S. *et al.* Unravelling the Mechanism of Ionic Fullerene Passivation for Efficient and Stable Methylammonium-free Perovskite Solar Cells. *ACS Energy Lett.* (2020).
- 13 Park, Y. H. *et al.* Inorganic rubidium cation as an enhancer for photovoltaic performance and moisture stability of HC(NH₂)₂PbI₃ perovskite solar cells. *Adv. Funct. Mater.* **27**, 1605988 (2017).
- 14 Stolterfoht, M. *et al.* Visualization and suppression of interfacial recombination for high-efficiency large-area pin perovskite solar cells. *Nat. Energy* **3**, 847-854 (2018).
- 15 Lee, J.-W. *et al.* 2D perovskite stabilized phase-pure formamidinium perovskite solar cells. *Nat. Commun.* **9**, 3021 (2018).
- 16 Lee, J.-W. *et al.* 2D perovskite stabilized phase-pure formamidinium perovskite solar cells. *Nat. Commun.* **9**, 1-10 (2018).
- 17 Min, H. *et al.* Efficient, stable solar cells by using inherent bandgap of α -phase formamidinium lead iodide. *Science* **366**, 749-753 (2019).
- 18 Prochowicz, D. *et al.* Engineering of perovskite materials based on formamidinium and cesium hybridization for high-efficiency solar cells. *Chem. of Mater.* **31**, 1620-1627 (2019).
- 19 Yadavalli, S. K. *et al.* Mechanisms of Exceptional Grain Growth and Stability in Formamidinium Lead Triiodide Thin Films for Perovskite Solar Cells. *Acta Materialia* (2020).
- 20 Kim, D. *et al.* Efficient, stable silicon tandem cells enabled by anion-engineered wide-bandgap perovskites. *Science* **368**, 155-160 (2020).
- 21 Han, T.-H. *et al.* Surface-2D/Bulk-3D Heterophased Perovskite Nanograins for Long-Term-Stable Light-Emitting Diodes. *Adv. Mater.* **32**, 1905674, doi:10.1002/adma.201905674 (2020).
- 22 Chen, Z. *et al.* Stable Sn/Pb-based perovskite solar cells with a coherent 2D/3D interface. *IScience* **9**, 337-346 (2018).

Reviewers' Comments:

Reviewer #1:

Remarks to the Author:

The authors have properly addressed all comments from reviewers. I am satisfied with the changes made in this revised manuscript. I recommend publication in its current form.

Reviewer #2:

Remarks to the Author:

The authors have addressed all the comments and concerns. The paper has been improved. In my opinion, it can be published now.